# Effects of the abacus-based mental calculation training application "SoroTouch" on cognitive functions: A randomized controlled trial

Tetsuya Takaoka[1]*, Keiji Hashimoto[1], Sayaka Aoki[2], Eisuke Inoue[3], Nobuyuki Kawate[1]

1 Department of Rehabilitation Medicine, Showa University School of Medicine, Tokyo, Japan, 2 Research Center for Advanced Science and Technology, The University of Tokyo, Tokyo, Japan, 3 Showa University Research Administration Center, Showa University, Tokyo, Japan

☯ These authors contributed equally to this work.
* ttakaoka@med.showa-u.ac.jp

**Data Availability Statement:** All the files can be accessed through the Dryad database. The following DOI has been assigned: doi:10.5061/dryad.1ns1rn8zx The dataset is also available by

## Abstract

This study investigated the effect of a home-based computerized cognitive training program that utilizes a digital application for training abacus-based mental calculations, "SoroTouch," on the cognitive functions of healthy middle-aged and older people using a randomized controlled trial. The participants were 20 adults (aged 42–79 years) who were involved in community-based activities for dementia prevention held by a certain organization. The participants were assigned randomly to the intervention (SoroTouch) group or control group. The SoroTouch group received home-based cognitive training with SoroTouch, being asked to use the software every day for 6 months, while the control group did not receive any intervention. To investigate the effect of SoroTouch, CogEvo, a cognitive functions test battery utilizing a tablet device, was administered to all participants once per month during the 6-month intervention period. In addition, before and after the intervention, all participants were asked to take the CogEvo and the Japanese version of the Montreal Cognitive Assessment (MoCA-J). The analyses showed that the SoroTouch group did not improve total scores of the CogEvo and MoCA-J, but large group differences were observed in the two tasks of the CogEvo as follows: 'Follow the order' (modified Trail Making Test) at 2 months after the beginning of the intervention (group differences; 39.4, 95% confidence interval; 7.6–71.2) and 'Route 99' at 6 months (group differences; 39.6, 95% confidence interval; 4.9–74.4). These results provide evidence that a home-based computerized cognitive training program SoroTouch has the potential to improve working memory, attention and planning in healthy middle-aged and older adults.

## Introduction

In Japan, the number of people aged 75 and over is increasing and expected to exceed 20 million by 2025 [1]. Compared to other countries' aging rates (the proportion of the population

clicking on the link below: https://doi.org/10.5061/dryad.1ns1rn8zx.

**Funding:** Digika Co.,Ltd. The funders had role in data collection about SoroTouch. They had no role in study design, analysis, decision to publish, or preparation of the manuscript.

**Competing interests:** Keiji Hashimoto is a corporate adviser to Total Brain Care CO., LTD. That company is the developer of the CogEvo. This does not alter to our adherence to PLOS ONE policies on sharing data and materials.

aged 65 and over), according to United Nations data, Japan's aging rate stands out at 29.1%, which is 5% ahead of second place, Italy (24.1%) [2]. Such an increase in the number of older people is considered to have a strong impact on our social system, due to an expected inflation in medical and nursing costs, which is called "the 2025 problem" in Japan [3]. In addition, since age increases the risk of cognitive decline, the number of people with cognitive decline is also expected to increase. Because cognitive decline, particularly dementia, affects individuals, their families, and even society, it is an important task to maintain and improve cognitive abilities of older adults. In such a societal context, technological digital tools that can maintain and improve the cognitive functions of middle-aged and older individuals are desired to aim for productivity and the maintenance and improvement of quality of life as a nation.

One of the strategies to improve cognitive functions in older adults is cognitive training [4, 5]. Cognitive training aims to train skills in specific cognitive domains with a standard set of tasks that are adjusted to an individual's level of ability. Cognitive training can be offered individually or in group sessions, and be held either in a paper and pencil format or a computerized format [4, 5]. Especially, home-based computerized cognitive training (CCT) has the advantage that it costs less and does not require a client to leave their home to attend a training session [6]. A systematic review conducted in 2014 indicated that group-based CCT programs have a positive effect on cognitive functions, such as memory, visuospatial abilities, and processing speed, for older adults without cognitive impairment [7]. While this review showed that home-based CCT programs are not effective, several studies with a randomized controlled trial design conducted after 2014 demonstrated a positive effect of home-based CCT [6, 8–15]. This means that home-based CCT can be a promising approach, specifically taking account of its feasibility. Since no evidence-based home-based CCT has been utilized widely in Japan, it is worthwhile to develop a new home-based CCT program for the older population of Japan and test its effect.

As for the content of such a home-based CCT program for older Japanese individuals, we considered that an abacus is a good candidate for the core content of the program. An abacus is a calculating tool that consists of movable beads on shafts that represent digits. The beads are manipulated following specific rules to perform calculations, such as addition and subtraction. In Japan, since people above a certain age are familiar with an abacus as they were exposed to this tool in elementary school education, cognitive training based on an abacus is expected to be easily accepted by this population. Since previous studies have shown cognitive training programs with a real abacus promote the cognitive functions of the trainees, such as visuospatial skills, working memory including central executive, and episodic and semantic memory [16–18], a home-based CCT program using an application that mimics an abacus may also be effective for improving cognitive functions in middle-aged and older Japanese people. In Japan, an application based on an abacus, Abacus-Based Mental Calculation Education Technology ("SoroTouch"), has been developed as an educational tool for children [19]. As 62% of the children who use the application reportedly acquire mental calculation abilities [19], SoroTouch is expected to have a positive effect on cognitive functions that is equivalent to the effect of a real abacus. Therefore, a home-based CCT program using this application may also improve the cognitive abilities of older adults.

The aim of this study was to investigate the effect of a home-based CCT program that utilizes SoroTouch on the cognitive functions of healthy middle-aged and older people. Based on the results of the study about CCT programs using a real abacus, we hypothesized that our home-based CCT program using a digital abacus would have a significant positive effect on the participants' visuospatial abilities and working memory.

## Materials and methods

### Study design

This study was a randomized controlled trial testing the effect of home-based CCT using Soro-Touch. As this is our first clinical trial of this application, we selected the office of the Niyo-katsu general incorporated association for the place to collect the primary outcome data which helped us recruit the study participants and let us use their space. Due to the availability of the human resource and physical space, the number of the participants was set as 20. With this sample size, a two-tailed test at a significance level of 0.05 can detect an effect size of 1.4 with a power of 0.8. The participants were assigned randomly to the intervention (SoroTouch) group or control group using the Mujinwari cloud service (https://mujinwari.biz/users/login). The SoroTouch group were asked to engage in cognitive training using SoroTouch every day for 6 months at home, while the control group did not receive any specific intervention. In order to investigate the effect of the intervention, all participants were administered tests of cognitive functions: CogEvo (Total Brain Care, Kobe, Japan) in the office of the Niyokatsu general incor-porated association and the Japanese version of the Montreal Cognitive Assessment (MoCA-J) [20] at a certain internist clinic, before and after the intervention. CogEvo was also adminis-tered to all participants once per month during the 6-month intervention period. In addition to these tests, before and after the intervention, the participants were also asked to fill in the 36-Item Short-Form Health Survey [21], a questionnaire measuring health-related quality of life, whose results are not reported in this paper since quality of life was not the focus of this study. The study design is illustrated in Fig 1. The intervention started in November 2021 and was completed in May 2022.

This study was approved by the Ethics Committee of Showa University on August 25, 2021. (Approval No. S7). All participants provided written informed consent prior to the administration of the first cognitive functions test, and received compensation of 5,000 yen per month for their participation, regardless of the assigned group. This study has been registered in the Japan Registry of Clinical Trials (JRCT), with the trial ID, jRCTs032210356.

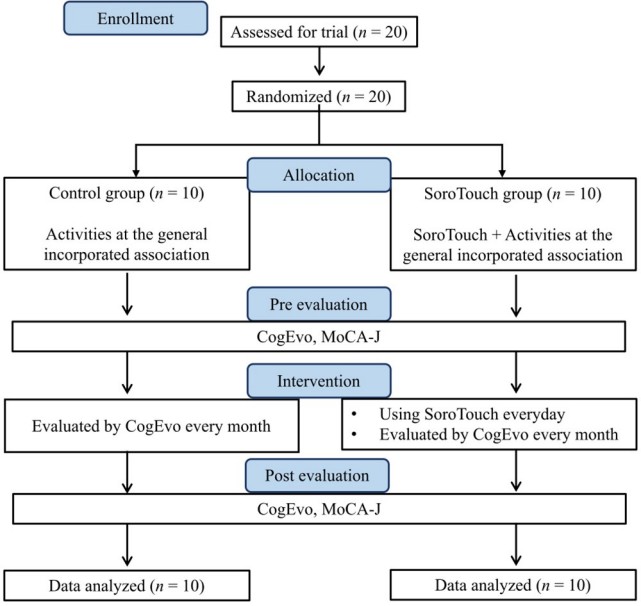

**Fig 1. Study design.**

**Table 1. Participants' demographic and clinical characteristics and their group comparison.**

| Characteristics | Control group | Soro Touch group |
|---|---|---|
| Sex | | |
| Male | 3 | 3 |
| Female | 7 | 7 |
| Age (years), mean (SD) | 64.4 (10.1) | 62.6 (12.7) |
| Education (years), mean (SD) | 14.2 (1.5) | 14.8 (1.7) |
| BMI (kg/m$^2$), means (SD) | 22.0 (2.5) | 22.1 (3.1) * |
| No exercise | 2 (20%) | 4 (40%) |
| No social contact | 0 (0%) | 1 (10%) |
| Current alcohol use | 6 (60%) | 3 (30%) |
| Current smoking | 2 (20%) | 3 (30%) |
| Past medical history, n (%) | | |
| Hypertension | 2 (20%) | 1 (10%) |
| Diabetes | 4 (40%) | 1 (10%) |
| Hearing impairment | 0 (0%) | 1 (10%) |
| Depression | 0 (0%) | 1 (10%) |
| Head injury | 0 (0%) | 1 (10%) |

Values are presented as means (SD) or *n* (%). No exercise refers to individuals who only exercise for less than 30 minutes per week; No social contact refers to individuals who socialize with people other than their family members less than once a week; Current alcohol use/smoking refers to individuals who habitually drink/smoke more than once a week. BMI, body mass index; NA, not applicable; SD, standard deviation.

*One person in the SoroTouch group did not report weight, so this is the average of 9 participants.

## Participants

The participants of this study were 20 individuals (6 men, 14 women) aged 42–79 years who were recruited from the participants of community-based activities for reducing the risk of dementia held by the Niyokatsu general incorporated association. The age range of the participants was selected for the following reasons; (1) In Japan, people aged 40 and over are legally insured by long-term care insurance, and have easier access to medical services for prevention of health problems, compared to younger people [22]. (2) A majority of individuals over 80 years old do not often use a computer or the internet, which prevents them from using the software without assistance [23]. The inclusion criteria were as follows: 1) age 40–79 years; and 2) willing to use SoroTouch for a period of 6 months. Individuals who met any of the following conditions were excluded: 1) diagnosed with dementia or mild cognitive impairment (MCI); and 2) unable to come to the office of the general incorporated association alone. The participants' demographic and clinical characteristics that are associated with the risk factors for dementia are shown in Table 1. This information was collected following the procedure presented by Livingston et al. [24].

## SoroTouch

SoroTouch (Fig 2) is an application software for tablet devices that trains individuals to acquire mental manipulation skills using a Japanese abacus, and it was originally developed for educational purposes. On this software, the user performs calculation exercises, such as $8 + 1 − 7 + 2 + 5$ and $29 + 13 + 57$, by touching and swiping ovals that resemble abacus beads on the screen, as well as by mental calculation. SoroTouch has an instructional section that teaches

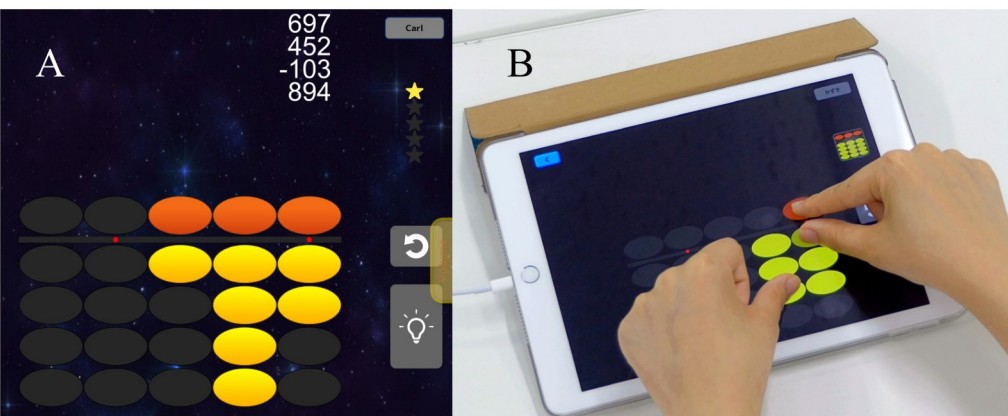

**Fig 2. SoroTouch.** (A) Display screen of SoroTouch. (B) A participant using SoroTouch.

participants how to perform calculations using the abacus. The task difficulty changes depending on the percentage of items answered correctly, but can also be controlled by the users themselves.

## CogEvo

CogEvo is a test battery of cognitive functions that assesses the function of various kinds of cognitive domains (visuospatial abilities, orientation, working memory, attention, planning, et al) through the performance of 12 cognitive tasks on a tablet device with a touch panel. In this study, we used the following five core tasks for measurement of the intervention outcomes (S1–S5 Files).

1. Same shape; The examinees are asked to select the same figure as the one displayed in the center of the panel from among six figures placed at different angles around (S1 File). The raw score of this task was found to be positively correlated with the score of Design copying in MMSE (r = 0.23, p<0.05) that reflects one's visuospatial abilities, in subjects of early stages of Alzheimer's Disease (AD) [25].

2. Orientation; This task requires the examinees to select the correct day, week and time of the examination day, and the 1–2 days before and after that day (S2 File). The performance of this task was positively correlated with the score of Temporal orientations in MMSE (r = 0.76, p<0.01) in subjects of early stages of AD [25].

3. Flashing light; The examinees are asked to remember the order in which the four colors of red, blue, green and yellow lights glow, and then, touch the lights in the same order. The score is calculated based on accuracy (S3 File). The score of this task was associated with the scores of Serial 7's of MMSE (r = 0.39, p<0.01) that reflects working memory in subjects of early stages of AD [25].

4. Route 99; The examinees are requested to trace squares on the panel from the starting point to the goal so that they pass randomly displayed digits in order from 1 to 10 (S4 File). This task requires arrangements to reach the goal quickly and measures planning that is one of the subdomains in executive functions [26]. The score of this task is known to be related to the scores of Serial 7's of MMSE (r = 0.43, p<0.01) that assesses one's working memory in subjects of early stages of AD [25].

5. Follow the order; This task is one version of Trail Making Test, which is one of the most frequently used neuropsychological tests that measures attention, specifically, an ability to sustain attention and an ability to shift attention [27–29]. On this task, the examinees are asked to touch the symbols displayed in the panel in a designated order. The type of symbols are numbers (1, 2, 3. . .), Japanese hiragana, and alphabet characters (A, B, C. . .), depending on the trial. In one trial, the examinee is asked to touch the numbers and alphabets alternatively (1, A, 2, B, 3, C. . .) (S5 File). The score of this task was positively correlated with the score of Serial 7's of MMSE (p = 0.41, p<0.01) in subjects of early stages of AD [25].

Sawada et al [30] previously demonstrated the validity and reliability of CogEvo and S. Ichii et al [31] reported that the tool can be used to evaluate age-related or pathological decline in cognitive functioning.

## MoCA-J

MoCA-J is a brief and useful face-to-face cognitive screening tool for MCI [20]. MoCA-J assesses abilities in the following eight cognitive domains: visuospatial abilities/executive functions, naming, attention, language, abstraction, delayed recall, and orientation. Naming, abstraction, delayed recall, and orientation are measured using a single task, and visuospatial abilities/executive functions, attention, and language are assessed according to performance in multiple tasks. Visuospatial abilities/executive functions are measured using the alternating trail making B, 3-D cube copy, and clock drawing tasks. Attention is measured using the digits forward and backward, vigilance sustained attention, and serial subtraction tasks. Language is assessed according to performance in the repetition and verbal fluency tasks. The total score of MoCA-J ranges from 0 to 30, and a cut-off score of 25/26 is used as an indicator of MCI. MoCA-J has shown high reliability and validity [20, 32].

## Intervention

The participants in the SoroTouch group were asked to use SoroTouch for 30 min every day for a period of 6 months at home. The status of the participants' usage (time they used the software for) and performance (whether they answered each item correctly) was monitored online by the staff of the general incorporated association. At the beginning of the intervention period, the participants in the SoroTouch group were provided a tablet device with SoroTouch installed and instructions on how to use the software in a face-to-face setting. Since one of the participants did not understand fully the initial instructions, she was provided with additional instructions via telephone approximately twice a month throughout the intervention period. At 1 month after the intervention started, one participant was found to use SoroTouch on only 13 days in the first month. Therefore, he received a phone call for encouragement twice a month so that he would use the software more frequently; however, he continued to use Soro-Touch for only approximately 10 days a month. The participants in the control group did not receive any specific cognitive training or other specific interventions.

## Data analysis

The primary outcomes were the total score and scores on each task of the CogEvo. Secondary outcomes included total scores and scores on each subcategory of the MoCA-J. To investigate the effect of SoroTouch, the participants' characteristics and scores for CogEvo and MoCA-J were compared between the SoroTouch and control groups. Because there were 6 primary outcomes and each was observed at six-time points, a Bonferroni correction was applied for our hypothesis testing. As a result, the adjusted criterion for statistical significance of the primary

outcomes was 0.0014, and *p*-values smaller than this threshold were considered significant. The descriptive statistics were reported using means and standard deviation (SD) for continuous variables. To analyze the effect of the intervention on the total scores and scores on each task of CogEvo, mixed effect models for repeated measures were applied. We had planned to use analysis of covariance, but we changed it to analyze repeated measures data before fixing clinical trial data. For the analyses, the baseline scores, groups (SoroTouch or control), timing of examination, and interaction between groups and timing of examination were set as covariates. To examine the effects of the intervention on the scores of MoCA-J, analysis of covariance was conducted, setting the baseline scores as covariates. We had planned to use an unpaired t-test, but we changed it to adjust for the baseline score. All statistical analyses were conducted using SPSS Statistics 28 (SPSS, Inc., Chicago, IL, USA).

## Results

The study enrolled 20 participants who were assigned randomly by a computer program to either the control group (*n* = 10; 3 men, 7 women) or SoroTouch group (*n* = 10; 3 men, 7 women) (Fig 1). Descriptive and clinical characteristics are shown in Table 1.

All participants completed all tests and questionnaires that measured the outcomes of the intervention at all time points of data collection, except for one participant in the control group who was unable to visit the office of the general incorporated association to take CogEvo at 3 months after the beginning of the intervention due to infection with coronavirus disease 2019. Except for this incident, no participants reported any adverse events that prevented them from continuing to participate in the study, such as developing dementia or any other diseases. During the 6-month intervention period, on average, the participants in the intervention group used SoroTouch on 158.7 days (SD: 35.4 days) and 24.7 min (SD: 9.2 min) per day. (See S1 Table for performance during the intervention for each participant in the SoroTouch group) One participant in the intervention group used SoroTouch for only 56 days in total and 8.4 min per day on average. This participant was included in our analysis since considerable variance in engagement in a training program among patients should be assumed in the clinical setting; thus, the presence of an individual who showed lower engagement was within expectations.

Table 2 displays the group differences in scores of the CogEvo between the SoroTouch group and the control group, as well as the results of the statistical comparisons between both groups. (See S2 Table for more detailed information, including mean values for both groups at each time point.) Fig 3 illustrates the amount of change from the baseline scores in the CogEvo within each group. Although scores of the CogEvo were not significantly different between the two groups, large group differences were observed in the score of the 'Follow the order' at 2 months (Fig 3E) and the score of the 'Route 99' at 6 months (Fig 3D), and in the scores of the 'Same shape' at 1 and 5 months. (Fig 3A).

Table 3 displays mean scores pre- and post-intervention and the difference in change between the two groups of the MoCA-J score. (See S3 Table for the amount of change in MoCA-J scores for both groups.) For MoCA-J, group difference was not observed in the change in the total and domain scores after the intervention.

## Discussion

To the best of our knowledge, the present study is the first randomized control trial in Japan to investigate the effect of an intervention with home-based CCT using software that trains mental calculations based on an abacus. The results of the present study suggested that training using SoroTouch did not improve total cognition measured with the CogEvo in healthy

**Table 2. The group differences in scores of the CogEvo between the SoroTouch group and the control group.**

| | | | Total | Same shape | Orientation | Flashing light | Route 99 | Follow the order |
|---|---|---|---|---|---|---|---|---|
| 1 month | Mean | | -23.4 | -69.8 | -13.7 | 41.2 | -19.5 | -8.5 |
| | 95% confidence interval | Lower bound | -332.4 | -127.1 | -80.1 | -147.0 | -68.3 | -50.8 |
| | | Upper bound | 285.6 | -12.6 | 52.8 | 229.3 | 29.3 | 33.8 |
| 2 months | Mean | | 49.5 | 0.9 | -57.6 | 43.9 | 24.7 | 39.4 |
| | 95% confidence interval | Lower bound | -116.8 | -62.3 | -134.1 | -62.0 | -13.6 | 7.6 |
| | | Upper bound | 215.8 | 64.0 | 18.8 | 149.8 | 63.1 | 71.2 |
| 3 months | Mean | | 3.2 | 14.8 | 12.4 | -50.6 | 37.1 | 23.2 |
| | 95% confidence interval | Lower bound | -209.1 | -53.7 | -50.5 | -184.5 | -4.5 | -6.7 |
| | | Upper bound | 215.5 | 83.3 | 75.3 | 83.4 | 78.8 | 53.1 |
| 4 months | Mean | | 4.8 | 16.1 | 32.3 | -28.5 | -14.3 | 28.8 |
| | 95% confidence interval | Lower bound | -118.8 | -32.1 | -20.3 | -113.4 | -63.0 | -1.7 |
| | | Upper bound | 128.5 | 64.3 | 84.9 | 56.4 | 34.3 | 59.4 |
| 5 months | Mean | | 4.7 | -108.7 | 30.6 | 27.3 | 19.6 | 6.1 |
| | 95% confidence interval | Lower bound | -120.0 | -175.1 | -0.9 | -92.6 | -22.8 | -28.0 |
| | | Upper bound | 129.4 | -42.3 | 62.2 | 147.1 | 62.0 | 40.1 |
| 6 months | Mean | | 55.9 | 28.9 | 4.3 | 23.8 | 39.6 | 19.4 |
| | 95% confidence interval | Lower bound | -173.1 | -23.7 | -37.0 | -86.1 | 4.9 | -2.1 |
| | | Upper bound | 285.0 | 81.6 | 45.6 | 133.6 | 74.4 | 40.8 |

The difference represents the value calculated by subtracting the score of the control group from the score of the SoroTouch group. Mixed effect models for repeated measures were applied to analyze. The baseline scores, groups (SoroTouch or control), timing of examination, and interaction between groups and timing of examination were set as covariates. A Bonferroni correction was applied for our hypothesis testing, and the adjusted criterion for statistical significance of the primary outcomes was 0.0014, and p-values smaller than this threshold were considered significant. The analysis showed no statistically significant differences in the scores between the two groups in either total or task scores.

middle-age and older adults. However, participants in the SoroTouch group had improved scores of working memory and attention at two months and planning at six months after the beginning of the intervention compared to the control group.

It was surprising that home-based CCT with the SoroTouch did not improve total cognitive functions. Interventions with a composite approach may be useful, rather than home-based CCT alone such as the present study. For example, in the Finnish Geriatric Intervention Study

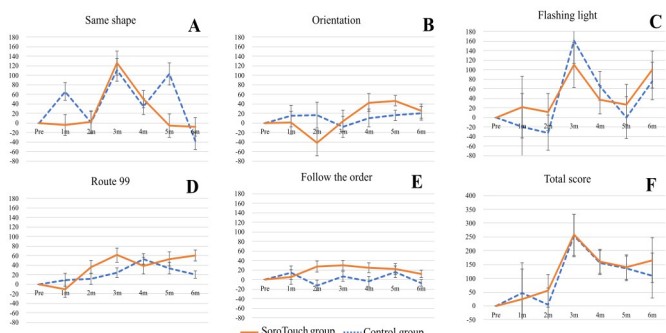

**Fig 3. Amount of change from baseline in the CogEvo scores within each group.** (A) Same shape. (B) Orientation. (C) Flashing light. (D) Route 99. (E) Follow the order. (F) Total score. Values are presented as means. Error bars indicate SE. X m, X months after the beginning of the intervention.

**Table 3. Mean MoCA-J scores pre- and post-intervention and the difference in change between the two groups.**

| | Control | | SoroTouch | | Difference between groups in change |
|---|---|---|---|---|---|
| | **Pre** | **Post** | **Pre** | **Post** | **Difference** |
| | **mean [95%CI]** | **mean [95%CI]** | **mean [95%CI]** | **mean [95%CI]** | **mean [95%CI]** |
| MoCA-J | | | | | |
| Total score | 27.3 [26.2; 28.4] | 29.1 [28.0; 30.2] | 27.7 [26.6; 28.8] | 29.8 [28.7; 30.9] | 0.3 [-1.2; 1.9] |
| Subcategory | | | | | |
| Visuospatial abilities / Executive function | 5.0 [4.9; 5.2] | 5.5 [5.4; 5.7] | 4.9 [4.8; 5.1] | 5.3 [5.2; 5.5] | -0.1 [-0.3; 0.1] |
| Alternating trail making | 1.0 [1.0; 1.0] | 1.5 [1.5; 1.5] | 1.0 [1.0; 1.0] | 1.5 [1.5; 1.5] | NA |
| Visuoconstruction skills (cube) | 1.0 [1.0; 1.0] | 1.5 [1.5; 1.5] | 1.0 [1.0; 1.0] | 1.5 [1.5; 1.5] | NA |
| Visuoconstruction skills (clock) | 3.0 [2.7; 3.3] | 3.5 [3.2; 3.8] | 2.9 [2.6; 3.2] | 3.3 [3.0; 3.6] | -0.1 [-0.3; 0.1] |
| Naming | 3.0 [3.0; 3.0] | 3.0 [3.0; 3.0] | 3.0 [3.0; 3.0] | 3.0 [3.0; 3.0] | NA |
| Attention | 4.9 [4.3; 5.5] | 4.8 [4.2; 5.4] | 5.5 [4.9; 6.1] | 5.9 [5.3; 6.5] | 0.6 [-0.4; 1.5] |
| Digit cpan | 1.7 [1.4; 2.0] | 1.9 [1.6; 2.2] | 1.8 [1.5; 2.1] | 2.2 [1.9; 2.4] | 0.1 [-0.2; 0.5] |
| Vigilance | 1.0 [1.0; 1.0] | 1.5 [1.5; 1.5] | 1.0 [1.0; 1.0] | 1.5 [1.5; 1.5] | NA |
| Serial sevens | 2.1 [1.6; 2.7] | 2.0 [1.4; 2.5] | 2.6 [2.0; 3.1] | 2.9 [2.4; 3.5] | 0.5 [-0.4; 1.3] |
| Language | 2.1 [1.7; 2.5] | 2.1 [1.7; 2.5] | 2.4 [2.0; 2.8] | 2.7 [2.3; 3.1] | 0.3 [-0.2; 0.8] |
| Sentence repetition | 1.2 [0.9; 1.4] | 1.1 [0.8; 1.4] | 1.4 [1.2; 1.7] | 1.6 [1.3; 1.9] | 0.3 [-0.1; 0.6] |
| Verbal fluency | 0.9 [0.7; 1.1] | 1.3 [1.2; 1.5] | 1.0 [0.8; 1.2] | 1.6 [1.4; 1.8] | 0.1 [-0.1; 0.3] |
| Abstraction | 1.7 [1.4; 2.0] | 2.0 [1.7; 2.3] | 1.8 [1.6; 2.1] | 2.2 [1.9; 2.4] | 0.1 [-0.2; 0.4] |
| Delayed recall | 4.5 [4.0; 5.0] | 6.0 [5.4; 6.5] | 3.9 [3.4; 4.4] | 4.7 [4.2; 5.3] | -0.6 [-1.4; 0.1] |
| Orientation | 6.0 [6.0; 6.0] | 6.5 [6.5; 6.5] | 6.0 [6.0; 6.0] | 6.5 [6.5; 6.5] | NA |

Analysis of covariance was conducted, setting the baseline scores as covariates. The difference in change between the groups is the change in the SoroTouch group minus the change in the Control group. The group difference was not observed in the change in the total and domain scores after the intervention. Some items could not be analyzed for the following reasons: 1) too few people changed; 2) the scores were the same for all; and 3) the scores were not a continuous number, e.g., only 0 or 1; these are indicated as not applicable (NA).

MoCA-J, The Japanese version of the Montreal Cognitive Assessment.

to Prevent Cognitive Impairment and Disability, a multidomain intervention program including cognitive training, diet, exercise, and vascular risk monitoring was found to improve or maintain the cognitive functions of older people [33]. A previous study has reported that a 12-month intervention with smartphone application-based Cognitive Training at Home improves total cognition in community-dwelling older individuals [6]. The study also included a 90-minute face-to-face cognitive intervention in groups of 5 to 10 people, monthly for 12 months, not home-based CCT alone. These trials imply that home-based CCT programs combined with face-to-face approaches may improve cognitive functions.

In the present study, modified TMT scores of the CogEvo in the SoroTouch group improved at 2 months after the beginning of the intervention, which may reflect the improvement in working memory and attention due to the short home-based CCT intervention. The CogEvo assesses cognitive functions in a similar way to the Trail Making Test, and the Trail Making Test-B is associated with working memory and executive functions including inhibitory control and working memory [34, 35]. It has been reported that a 3-week intervention with a home-based complex verbal working memory training program improved working memory in healthy individuals over 60 years old [36], and that a 5-week home-based CCT improved working memory in cognitively normal adults [14]. These suggest that home-based CCT programs improve working memory after a short intervention lasting from a few weeks to two months, but that the effects do not continue for longer periods of time. On the other

hand, it is possible that participants' number skills improved rather than their working memory, since "Route99" and "Follow the Order" were similar to SoroTouch in their use of numbers.

In this study, one result that was somewhat confusing was that visuospatial ability improved more in the control group than in the SoroTouch group. The lack of an improvement in this ability in the SoroTouch group was surprising since mental manipulation using an abacus requires the mental manipulation of an "image" of an abacus, and so training this skill is expected to enhance visuospatial recognition. One possible factor that affected the result is the reliability of the assessment tool CogEvo. While CogEvo measures total cognition, attention, memory, and executive functions with fair-to-good reliability [37], the score on the task assessing visuospatial recognition, 'Same Shape', does not show high reliability. This means that the domain scores obtained in this study might not accurately reflect the participants' visuospatial ability. To investigate the effect of SoroTouch on visuospatial ability, further studies are necessary using a different evaluation tool that is more reliable in this domain.

In addition to the findings above, it is worthwhile highlighting the fact that home-based CCT using SoroTouch was quite feasible. The participants in the SoroTouch group used the software on more than 85% of the days in the 6-month intervention period, for an average of approximately 25 min/day, even though they were not monitored directly by others. This observation shows that the participants did not feel too much of a burden when using SoroTouch. This further indicates the possibility that clinicians can prescribe SoroTouch as an easy-to-administer tool for improving cognitive functions that can be used anywhere, any time. Since Japan has many older people and some of them have difficulty leaving home due to their health and physical condition, the development of such a tool is quite important to maintain their mental health, which could be the most significant contribution made by the present study.

## Study limitations

This study has several limitations that merit consideration. First, the participants were not followed-up after the intervention period was over; therefore, no information was obtained about how long the effect of the intervention lasted for. Second, the present study had 10 participants each in the control and SoroTouch groups, and the small sample size resulted in low power. Third, all participants were healthy and recruited from a group engaging in community activities provided by a specific organization. We aimed to adapt home-based cognitive training with the SoroTouch for use in the general community, specifically for older individuals who are cognitively healthy, so we consider our findings are applicable to a broader population. However, further trials are needed to investigate the effectiveness of home-based CCT with the SoroTouch for people at risk of MCI and cognitive impairment. Hence, larger and longer studies should be planned with more diverse participants including who have already started to develop dementia or MCI. Lastly, we were unable to confirm the identity of the actual users of SoroTouch, like other home-based online activities without a video, though we were able to track the exact timing of each participant's usage daily as they used SoroTouch online. To address these limitations, regular follow-up sessions, either in person or online, should be included in the protocol for future trials.

## Conclusions

The home-based intervention with the abacus-based mental calculation application Soro-Touch did not improve total cognition in healthy middle-age and older people. However, these results of the present study provided evidence that a home-based computerized cognitive

training program SoroTouch has the potential to improve working memory, attention and planning in healthy middle-aged and older adults.

## Supporting information

**S1 File. Same shape.** This is a video of the CogEvo task 'Same shape'. It provides only part of the task, not all of it.
(MOV)

**S2 File. Orientation.** This is a video of the CogEvo task 'Orientation'. It provides only part of the task, not all of it.
(MOV)

**S3 File. Flashing light.** This is a video of the CogEvo task 'Flashing light'. It provides only part of the task, not all of it.
(MOV)

**S4 File. Route 99.** This is a video of the CogEvo task 'Route 99'. It provides only part of the task, not all of it.
(MOV)

**S5 File. Follow the order.** This is a video of the CogEvo task 'Follow the order'. It provides only part of the task, not all of it.
(MOV)

**S1 Table. Performance during the intervention for each participant in the SoroTouch group.** "Usage time" represents the time (in minutes) that the participant performed Soro-Touch during the intervention period."Number of answers" represents the number of questions solved by the participant during the intervention period. "Number of correct answers" represents the number of questions solved by the participant that were answered correctly. "Number of wrong answers" represents the number of questions that the participant solved incorrectly. "Final level of achievement" is the level of mental arithmetic that the participant was able to perform after 6 months of intervention. For example, "Addition and subtraction of three 2-digit units." represents addition or subtraction such as "52+28–30." "Addition and subtraction of two 2-digit units and one 1-digit unit." represents addition or subtraction such as "67+4+23." "1-digit × 2-digit multiplication" represents multiplication such as "23 x 6," and "4-digit ÷ 1-digit division" represents division such as "8685 ÷ 3".
(XLSX)

**S2 Table. Mean scores of the CogEvo and differences between the two groups.** The Difference represents the value calculated by subtracting the score of the control group from the score of the SoroTouch group. Mixed effect models for repeated measures were applied to analyze. The baseline scores, groups (SoroTouch or control), timing of examination, and interaction between groups and timing of examination were set as covariates. A Bonferroni correction was applied for our hypothesis testing, and the adjusted criterion for statistical significance of the primary outcomes was 0.0014, and p-values smaller than this threshold were considered significant. The analysis showed no statistically significant differences in the scores between the two groups in either total or task scores.
(XLSX)

**S3 Table. Baseline scores, the amount of changes and group differences in change of the MoCA-J score.** The difference in change between the groups is the change in the SoroTouch group minus the change in the Control group. Analysis of covariance was conducted, setting

the baseline scores as covariates. The group difference was not observed in the change in the total and domain scores after the intervention. Some items could not be analyzed for the following reasons: 1) too few people changed; 2) the scores were the same for all; and 3) the scores were not a continuous number, e.g., only 0 or 1; these are indicated as not applicable (NA). MoCA-J, The Japanese version of the Montreal Cognitive Assessment.
(XLSX)

**S1 Checklist. CONSORT 2010 checklist of information to include when reporting a randomised trial\*.**
(DOC)

**S1 Data.**
(DOCX)

## Acknowledgments

The authors would like to thank the participants who cooperated with the research and members of the General Incorporated Association, Niyokatsu (Dementia prevention).

## Author Contributions

**Conceptualization:** Keiji Hashimoto.

**Data curation:** Tetsuya Takaoka.

**Formal analysis:** Eisuke Inoue.

**Funding acquisition:** Keiji Hashimoto.

**Investigation:** Tetsuya Takaoka.

**Methodology:** Keiji Hashimoto, Sayaka Aoki, Eisuke Inoue.

**Project administration:** Keiji Hashimoto.

**Supervision:** Keiji Hashimoto, Nobuyuki Kawate.

**Validation:** Keiji Hashimoto.

**Visualization:** Tetsuya Takaoka.

**Writing – original draft:** Tetsuya Takaoka.

**Writing – review & editing:** Keiji Hashimoto, Sayaka Aoki, Eisuke Inoue, Nobuyuki Kawate.

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
