## [Decision Letter · Decision Letter 0]

29 Mar 2023

PONE-D-23-01459Effects of the abacus-based mental calculation training application “SoroTouch” on cognitive function: a randomized controlled trialPLOS ONE

Dear Dr. Takaoka,

Thank you for submitting your manuscript to PLOS ONE. After careful consideration, we feel that it has merit but does not fully meet PLOS ONE’s publication criteria as it currently stands. Therefore, we invite you to submit a revised version of the manuscript that addresses the points raised during the review process.

We look forward to receiving your revised manuscript.

Kind regards,

Eshak I Bahbah

Academic Editor

PLOS ONE

“This study funded by Digika Co.,Ltd. The funder was involved in data collection.”

“Keiji Hashimoto is a corporate adviser to Total Brain Care CO., LTD. That company is the developer of CogEvo.”

Please respond by return email with your amended Competing Interests Statement and we will change the online submission form on your behalf.

Reviewers' comments:

Reviewer's Responses to Questions

**Comments to the Author**

1. Is the manuscript technically sound, and do the data support the conclusions?

Reviewer #1: No

Reviewer #2: Yes

Reviewer #3: Yes

2. Has the statistical analysis been performed appropriately and rigorously? 

Reviewer #1: No

Reviewer #2: Yes

Reviewer #3: No

3. Have the authors made all data underlying the findings in their manuscript fully available?

Reviewer #1: No

Reviewer #2: Yes

Reviewer #3: Yes

4. Is the manuscript presented in an intelligible fashion and written in standard English?

Reviewer #1: Yes

Reviewer #2: Yes

Reviewer #3: Yes

5. Review Comments to the Author

Reviewer #1: This study uses a home-based abacus training program to train middle aged to elderly individuals, with a 6 month follow up, for improving cognitive functioning. The results show very mild training effect for Sorotech. However, because the statistics are not performed properly, as of now I am not quite convinced that the training effects are there. Comments below.

1. The repeated emphasis of dementia is not necessary because all the participants here are healthy adults, and people with dementia were actually excluded. As such, this is more of a training program for healthy adults. To link this with dementia (or the prevention of it) is too far fetched at the moment.

2. This study is characterized as a clinical trial, therefore the trial number from https://clinicaltrials.gov/ should be provided

3. CogEvo is used as the evaluation tool, but how exactly does it test visuospatial cognition, orientation, memory, executive function, and attention? From my experience, most market softwares that claims ot test these cognitive functions usually blend a lot of unnecessary components into tasks, such that (for example) executive function is conflated with attention, or visuospatial cognition confounded with attention…etc. Therefore, I think the current study is testing a novel training program, and testing it with yet another novel program, so the training effect (or lack thereof) becomes difficult to explain because the cognitive/psychological construct behind each training/testing is unclear.

4. Finally and most importantly, it seems that the authors have not done ANOVA or multiple comparison correction of any kind in the results. There are too many t-tests going on, which inflates the chances for false positives. The authors should first perform an ANOVA (with training vs. control, time…etc as factors), and if significant, then move on to posthoc t-tests with (preferably) Bonferroni correction or bonferroni holm correction. Only contrasts that survive ANOVA and posthoc multiple comparison correction should be reported in Results section.

Reviewer #2: Major issues:

1.Throughout the manuscript authors discuss about dementia and preventing dementia. Dementia can be caused by several diseases which all lead to damage in the brain. It is true that several lifestyle factors increase the risk of cognitive decline, however in the case of for example Alzheimer’s disease there is also factors that can not be treated or prevented. Additionally, it is unlikely (unfortunately) that cognitive training without any additional lifestyle intervention would prevent dementia. In the case of this study, in my opinion, it would be better state that computer-based cognitive training may maintain or improve cognitive functions rather than prevent dementia.

2. There is no rationale why authors recruited participants who were at age of 40 to 79. And why they choose cognitively healthy individuals (in study protocol the objectives of the study state that in the future it is expected that SoroTouch will prevent dementia in patients with MCI). Were they at risk of developing cognitive decline?

3. Authors do not provide enough information about participant characteristics (for example, what is the definition of no exercise), training adherence and the way it was monitored and how CogEvo assess the cognitive domains (for example, executive functions is an umbrella term for several sub-domains). Concerning the training adherence how was taken care to ensure that no family member used the exercise program instead of the participant?

4. Sample size in this study was very small and according to Table 1 they were not at risk for cognitive impairment (they were educated, they had normal weight, they exercised, did not use alcohol or smoked and they have almost non chronic diseases that are associated with increased risk of cognitive decline). Therefore, I think that the results of this study can not be generalized outside of this sample.

Minor issues

1. Please use “older adults” instead of elderly

2. Executive functions is an umbrella term for several subdomains and therefore it should be plural.

3. Visuospatial abilities instead of visuospatial cognition

4. Why there is that much missing information in table 2? In this table we unfortunately can see the challenges of small sample size.

Reviewer #3: Dear Editor,

Thank you for the opportunity to provide a review of Manuscript PONE-D-23-01459 entitled "EEffects of the abacus-based mental calculation training application “SoroTouch” on cognitive function: a randomized controlled trial". My comments relate primarily to the adequacy of the implementation and reporting of epidemiologic and statistical procedures.

The quality of the technical English is appropriate and offered no bar to my evaluation of the manuscript.

# Major Issues

## Abstract

The authors need to quantify the results. They must avoid qualitative statements of effect such as one group experiencing "significantly greater improvement" compared to another. They must report the measure of effect and apply a confidence interval.

## Methods

The authors must identify the primary outcome and all secondary outcomes.

The authors fail to report any safety or adverse events outcomes. Why is this missing?

The authors must report their calculations for sample size.

The authors must report standard deviations, not standard errors. The use of standard errors is highly inappropriate.

## Results

The authors MUST NOT apply formal statistical testing between the groups in Table 1. This is highly inappropriate because it sets up a statistical tautology. The last column must be deleted, as well as the last sentence of the footnote.

Figure 3 is a table and should not be presented as a figure.

In Figure 4, the position of the labels for the horizontal axis is variable. In addition, the scale of the vertical axis is highly variable. These two components of the graph must be fixed for all graphs.

Thank you.

6. PLOS authors have the option to publish the peer review history of their article (what does this mean?). If published, this will include your full peer review and any attached files.

Reviewer #1: No

Reviewer #2: No

Reviewer #3: No

---

## [Author Response · Author response to Decision Letter 0]

9 May 2023

Thank you very much for your thorough comments. As you pointed out, because of its design, this study cannot directly test whether the abacus-based mental calculation training application prevents older people from suffering from mild cognitive impairment or dementia. Nevertheless, in Japan, where a super-aging society is emerging, it is crucial to maintain the cognitive function of healthy middle-aged and older individuals as much as possible. Thus, the purpose of this study was restated as investigation of the effect of an abacus-based mental calculation training application on 

the cognitive function of healthy middle-aged and elderly individuals.

Reviewer #1: This study uses a home-based abacus training program to train middle aged to elderly individuals, with a 6 month follow up, for improving cognitive functioning. The results show very mild training effect for Sorotech. However, because the statistics are not performed properly, as of now I am not quite convinced that the training effects are there. Comments below.

Response: Thank you for your careful peer review. We have made all possible corrections and would appreciate another peer review.

1. The repeated emphasis of dementia is not necessary because all the participants here are healthy adults, and people with dementia were actually excluded. As such, this is more of a training program for healthy adults. To link this with dementia (or the prevention of it) is too far fetched at the moment.

Response: As you have pointed out, this study was not conducted on patients with dementia or mild cognitive impairment, but on healthy subjects. Therefore, the text has been substantially rewritten as a study that investigated the effect of a cognitive training application on healthy middle-aged and elderly subjects.

See the section of Introduction in p.3-5.

2. This study is characterized as a clinical trial, therefore the trial number from https://clinicaltrials.gov/ should be provided.

Response: Our study is not registered in https://clinicaltrials.gov/, but in the Japan Registry of Clinical Trials (JRCT). The trial ID can be obtained in their website, “https://jrct.niph.go.jp/en-latest-detail/jRCTs032210356.” The information was also added in the last paragraph in the Study Design section (p.7). 

3. CogEvo is used as the evaluation tool, but how exactly does it test visuospatial cognition, orientation, memory, executive function, and attention? From my experience, most market softwares that claims ot test these cognitive functions usually blend a lot of unnecessary components into tasks, such that (for example) executive function is conflated with attention, or visuospatial cognition confounded with attention…etc. Therefore, I think the current study is testing a novel training program, and testing it with yet another novel program, so the training effect (or lack thereof) becomes difficult to explain because the cognitive/psychological construct behind each training/testing is unclear.

Response: Thank you for your appropriate comment. Like any other assessment tool, CogEvo is unable to measure one specific cognitive ability without being influenced by other cognitive abilities. However, it is also true that a previous study showed that the item scores of CogEvo are significantly associated with the those of the MMSE, which was designed to measure one’s visuospatial abilities, orientation, working memory, and executive function. Therefore, we decided to describe the outcome of the intervention using the item scores of the CogEvo, not the name of the ability each item is supposed to measure, and add our “interpretation” in the Discussion section. We also provided a task description of each item of the CogEvo more in detail and the strengths of the correlation between the item scores of the CogEvo and those of the MMSE found in a previous study in the CogEvo subsection. Moreover, we also added a video clip to visually demonstrate how an examinee is engaged in each task, as supporting information. 

4. Finally and most importantly, it seems that the authors have not done ANOVA or multiple comparison correction of any kind in the results. There are too many t-tests going on, which inflates the chances for false positives. The authors should first perform an ANOVA (with training vs. control, time…etc as factors), and if significant, then move on to posthoc t-tests with (preferably) Bonferroni correction or bonferroni holm correction. Only contrasts that survive ANOVA and posthoc multiple comparison correction should be reported in Results section.

Response: Since a mixed-effects model for repeated measures (MMRM) was used to analyze CogEvo, standard ANOVA could not be applied. Therefore, only the primary endpoint, the total CogEvo score at 6 months, was evaluated for statistical significance, while other evaluations including CogEvo at other time points and secondary endpoints were considered exploratory analyses to avoid multiple comparisons. In exploratory analyses, only p-values were presented without stating whether they were statistically significant or not ("significant" or "not significant" was not mentioned in the manuscript). See the section of Study design in P5-6.

Reviewer #2: Major issues:

1. Throughout the manuscript authors discuss about dementia and preventing dementia. Dementia can be caused by several diseases which all lead to damage in the brain. It is true that several lifestyle factors increase the risk of cognitive decline, however in the case of for example Alzheimer’s disease there is also factors that can not be treated or prevented. Additionally, it is unlikely (unfortunately) that cognitive training without any additional lifestyle intervention would prevent dementia. In the case of this study, in my opinion, it would be better state that computer-based cognitive training may maintain or improve cognitive functions rather than prevent dementia.

Response: Thank you for pointing this out. Referring to your opinion, we have substantially reframed the study as one about whether computer-based cognitive training has a potential to maintain or improve cognitive function in healthy individuals, rather than prevention of dementia (see the Introduction, p.3-5). 

 2. There is no rationale why authors recruited participants who were at age of 40 to 79. And why they choose cognitively healthy individuals (in study protocol the objectives of the study state that in the future it is expected that SoroTouch will prevent dementia in patients with MCI). Were they at risk of developing cognitive decline?

Response: Thank you for your appropriate comments. In this updated version of the manuscript, we now clearly stated the reason that this study selected people aged 40 to 79 years old as its participants. With regard to the cognitive status of the participants, they do not report any specific risk factors of cognitive decline. However, it is no longer problematic, because we restated the purpose of the study as an investigation of whether SoroTouch improves cognitive function in healthy middle-aged people, taking account of the actual study protocol.

See the section of Participants in P7.

3. Authors do not provide enough information about participant characteristics (for example, what is the definition of no exercise), training adherence and the way it was monitored and how CogEvo assess the cognitive domains (for example, executive functions is an umbrella term for several sub-domains). Concerning the training adherence how was taken care to ensure that no family member used the exercise program instead of the participant?

Response: In response to your comment, we added definitions for each item regarding participant characteristics in the legends in Table 1. Regarding the training adherence, because the participants used SoroTouch online, we were able to know the exact time when Sorotouch was used on each day, but not to confirm who actually used the application, like any home-based online activity without a video-recording. Since this fact should be recognized as one of the limitations of this study, we added it in the Study Limitations in P20.

As for the cognitive domains assessed using the CogEvo, we now added description of each task more in detail and information about the relationship between the item scores of the CogEvo and those of the MMSE in the CogEvo section. Finally, to illustrate the procedure of cognitive assessment in the CogEvomore clearly, we added a video in which an examinee uses CogEvo as supporting information (S1-5 Files).

4. Sample size in this study was very small and according to Table 1 they were not at risk for cognitive impairment (they were educated, they had normal weight, they exercised, did not use alcohol or smoked and they have almost non chronic diseases that are associated with increased risk of cognitive decline). Therefore, I think that the results of this study can not be generalized outside of this sample.

Response: As mentioned in the response to your other comment and at the beginning of this letter, this study is now reframed as a study that investigates the effect of Solotouch on healthy middle-aged adults. Nevertheless, it is true that the small sample size reduces the generalizability of this study. Therefore, we added this concern in the Limitation section (see p.20).

Minor issues

1. Please use “older adults” instead of elderly.

2. Executive functions is an umbrella term for several subdomains and therefore it should be plural.

3. Visuospatial abilities instead of visuospatial cognition.

Response: We have rephrased the terms following your suggestions.

4. Why there is that much missing information in table 2? In this table we unfortunately can see the challenges of small sample size.

Response: As stated in Table 2 legend, some items could not be analyzed for the following reasons: 1) too few people changed; 2) the scores were the same for all; and 3) the scores were not a continuous number, e.g., only 0 or 1. In addition, as mentioned above, this was an exploratory study so we added our understanding that larger sample size trials will need to be conducted in the future, in the Discussion.

Reviewer #3: 

Thank you for the opportunity to provide a review of Manuscript PONE-D-23-01459 entitled "EEffects of the abacus-based mental calculation training application “SoroTouch” on cognitive function: a randomized controlled trial". My comments relate primarily to the adequacy of the implementation and reporting of epidemiologic and statistical procedures.

The quality of the technical English is appropriate and offered no bar to my evaluation of the manuscript.

# Major Issues

## Abstract

The authors need to quantify the results. They must avoid qualitative statements of effect such as one group experiencing "significantly greater improvement" compared to another. They must report the measure of effect and apply a confidence interval.

Response: We have removed the qualitative statements and added confidence intervals in Abstract in P2.

## Methods

The authors must identify the primary outcome and all secondary outcomes.

Response: Primary and secondary outcomes were identified and described in the Data analysis in P12 as follows; The primary outcome measure was the total scores of the CogEvo at 6 months after the beginning of the intervention. Secondary outcomes included total scores of the CogEvo at other time points, scores on each subcategory of the CogEvo from 1 to 6 months after the beginning of the intervention, and total scores and scores on each subcategory of the MoCA-J.

The authors fail to report any safety or adverse events outcomes. Why is this missing?

Response: As stated in the Results, one participant contracted COVID-19 unrelated to the present study. Except for this incident, no participants reported any adverse events that prevented them from continuing to participate in the study, such as developing dementia or any other diseases. (Please see the second paragraph of the Result in P15as follows; All participants completed all tests and questionnaires that measured the outcomes of the intervention at all time points of data collection, except for one participant in the control group who was unable to visit the office of the general incorporated association to take CogEvo at 3 months after the beginning of the intervention due to infection with coronavirus disease 2019. )

The authors must report their calculations for sample size.

Response: We have reported the following in Study design section in P5-6 about the sample size calculations as follows; Based on the results of the present study on cognitive function and quality of life, a larger-scale validation study is planned in the future. Therefore, this study was planned as an exploratory study. The number of patients was set at 20, the maximum number of patients that could be conducted at the relevant institution. With this sample size, a two-tailed test at a significance level of 0.05 can detect an effect size of 1.4 with a power of 0.8. 

The authors must report standard deviations, not standard errors. The use of standard errors is highly inappropriate.

Response: We reported SD in Table 1. Others reported 95% confidence intervals.

## Results

The authors MUST NOT apply formal statistical testing between the groups in Table 1. This is highly inappropriate because it sets up a statistical tautology. The last column must be deleted, as well as the last sentence of the footnote.

Response: We deleted the last column in Table 1 in P8 as you have suggested.

Figure 3 is a table and should not be presented as a figure.

Response: Figure 3 has been modified as a table.

In Figure 4, the position of the labels for the horizontal axis is variable. In addition, the scale of the vertical axis is highly variable. These two components of the graph must be fixed for all graphs.

Response: As indicated, the position of the labels for the horizontal axis and the scale for the vertical axis have been fixed.

---

## [Decision Letter · Decision Letter 1]

8 Jun 2023

PONE-D-23-01459R1Effects of the abacus-based mental calculation training application “SoroTouch” on cognitive function: a randomized controlled trialPLOS ONE

Dear Dr. Takaoka,

Thank you for submitting your manuscript to PLOS ONE. After careful consideration, we feel that it has merit but does not fully meet PLOS ONE’s publication criteria as it currently stands. Therefore, we invite you to submit a revised version of the manuscript that addresses the points raised during the review process.

ACADEMIC EDITOR:Authors must respond to the comments of both reviewers and consider performing multiple comparison corrections using the Bonferroni test.Please submit your revised manuscript by Jul 23 2023 11:59PM. If you will need more time than this to complete your revisions, please reply to this message or contact the journal office at plosone@plos.org. Please include the following items when submitting your revised manuscript:A rebuttal letter that responds to each point raised by the academic editor and reviewer(s). You should upload this letter as a separate file labeled 'Response to Reviewers'.A marked-up copy of your manuscript that highlights changes made to the original version. You should upload this as a separate file labeled 'Revised Manuscript with Track Changes'.An unmarked version of your revised paper without tracked changes. You should upload this as a separate file labeled 'Manuscript'.

We look forward to receiving your revised manuscript.

Kind regards,

Eshak I Bahbah

Academic Editor

PLOS ONE

Additional Editor Comments:

Authors must respond to the comments of both reviewers and consider performing multiple comparison corrections using the Bonferroni test.

Reviewers' comments:

Reviewer's Responses to Questions

**Comments to the Author**

1. If the authors have adequately addressed your comments raised in a previous round of review and you feel that this manuscript is now acceptable for publication, you may indicate that here to bypass the “Comments to the Author” section, enter your conflict of interest statement in the “Confidential to Editor” section, and submit your "Accept" recommendation.

Reviewer #1: (No Response)

Reviewer #2: (No Response)

2. Is the manuscript technically sound, and do the data support the conclusions?

Reviewer #1: No

Reviewer #2: Yes

3. Has the statistical analysis been performed appropriately and rigorously? 

Reviewer #1: No

Reviewer #2: Yes

4. Have the authors made all data underlying the findings in their manuscript fully available?

Reviewer #1: No

Reviewer #2: Yes

5. Is the manuscript presented in an intelligible fashion and written in standard English?

Reviewer #1: Yes

Reviewer #2: Yes

6. Review Comments to the Author

Reviewer #1: I appreciate the efforts that the authors have made to address my comments. The introduction now looks okay. However, my most important comment about statistics was ignored by the authors. Multiple comparison or family wise error rate protection should be done, otherwise there is the risk of inflated error rate and hence high false positives. Right now the authors simply claims that everything is exploratory analysis, and still go ahead with an asterisk for p<0.05. I do not agree with the use of this framing to evade multiple comparison corrections. I recommend the authors to use Bonferroni correction, or Bonferroni-holm if Bonferroni is too strict.

Reviewer #2: I want to thank the authors for the revised manuscript. I feel that they have successfully improved the manuscript. However, I have still some suggestions how to improve the manuscript.

1. Cognitive functions and executive functions should be plural throughout the manuscript.

2. Reference group should be added in tables that investigate differences between the groups

3. I still feel that it is little bit problematic how executive functions are described in the manuscript. Executive functions is an umbrella term for several subdomains including inhibition, set-shifting and working memory updating. Additionally, Trail Making Test assess some but not all subdomains of executive functions.

7. PLOS authors have the option to publish the peer review history of their article (what does this mean?). If published, this will include your full peer review and any attached files.

Reviewer #1: No

Reviewer #2: No

---

## [Author Response · Author response to Decision Letter 1]

3 Jul 2023

We are very grateful for your careful review. Here are the responses to your comments. 

Reviewer #1: I appreciate the efforts that the authors have made to address my comments. The introduction now looks okay. However, my most important comment about statistics was ignored by the authors. Multiple comparison or family wise error rate protection should be done, otherwise there is the risk of inflated error rate and hence high false positives. Right now the authors simply claims that everything is exploratory analysis, and still go ahead with an asterisk for p<0.05. I do not agree with the use of this framing to evade multiple comparison corrections. I recommend the authors to use Bonferroni correction, or Bonferroni-holm if Bonferroni is too strict.

Response: Following the recommendations of you and the editor, we applied a Bonferroni correction for our hypothesis testing. As a result, the adjusted criterion for statistical significance of the primary outcome was found to be 0.0014, and p-values smaller than this threshold were considered significant. Referring to the new results obtained after the adjustment, we changed the description in the Data analysis and Results section, Table 2, and Figure 3. (See p.12, 14-16, respectively)

Reviewer #2: I want to thank the authors for the revised manuscript. I feel that they have successfully improved the manuscript. However, I have still some suggestions how to improve the manuscript.

1. Cognitive functions and executive functions should be plural throughout the manuscript.

Response: We have replaced the terms following your suggestions.

2. Reference group should be added in tables that investigate differences between the groups

Response: Due to the space limit, we put just “difference in means” between the 2 groups for each variable at each timing in Tables 2 and Table 3, and actual mean scores and confidence intervals for each group in Tables S1 and S2. If you think it is necessary to provide the mean scores and SD of both SoroTouch group and the Control group in Tables 2 and 3, please let us know. 

3. I still feel that it is little bit problematic how executive functions are described in the manuscript. Executive functions is an umbrella term for several subdomains including inhibition, set-shifting and working memory updating. Additionally, Trail Making Test assess some but not all subdomains of executive functions.

Response: Thank you very much for giving us a comment that helps us consider the concepts more thoroughly. After consideration, we decided not to use the term, "executive functions," but directly name the subdomains instead when describing the methods and results of our study. However, we remained to use the term, “executive functions” when we refer to previous studies which adopted the term. As for the Trail Making Test, we rewrote its description to clarify what subdomain of executive function it actually measures (See the section, CogEvo, p.10). In a similar way, we modified the description of the subtest, “Follow the Order” and “Route 99.” 

In addition to the revisions above, we also made the data we analyzed available to follow the PLOS data policy. The data are available on Dryad, https://datadryad.org/stash. The uploaded dataset included almost all the demographic and clinical characteristics of the participants and the CogEvo and MoCA-J scores. We recoded the age so that the data becomes not personally identifiable.

---

## [Decision Letter · Decision Letter 2]

30 Aug 2023

PONE-D-23-01459R2Effects of the abacus-based mental calculation training application “SoroTouch” on cognitive functions: a randomized controlled trialPLOS ONE

Dear Dr. Takaoka,

Thank you for submitting your manuscript to PLOS ONE. After careful consideration, we feel that it has merit but does not fully meet PLOS ONE’s publication criteria as it currently stands. Therefore, we invite you to submit a revised version of the manuscript that addresses the points raised during the review process.

We look forward to receiving your revised manuscript.

Kind regards,

Eshak I Bahbah

Academic Editor

PLOS ONE

Reviewers' comments:

Reviewer's Responses to Questions

**Comments to the Author**

1. If the authors have adequately addressed your comments raised in a previous round of review and you feel that this manuscript is now acceptable for publication, you may indicate that here to bypass the “Comments to the Author” section, enter your conflict of interest statement in the “Confidential to Editor” section, and submit your "Accept" recommendation.

Reviewer #4: All comments have been addressed

Reviewer #5: (No Response)

Reviewer #6: (No Response)

2. Is the manuscript technically sound, and do the data support the conclusions?

Reviewer #4: Yes

Reviewer #5: No

Reviewer #6: Yes

3. Has the statistical analysis been performed appropriately and rigorously? 

Reviewer #4: Yes

Reviewer #5: No

Reviewer #6: No

4. Have the authors made all data underlying the findings in their manuscript fully available?

Reviewer #4: Yes

Reviewer #5: Yes

Reviewer #6: Yes

5. Is the manuscript presented in an intelligible fashion and written in standard English?

Reviewer #4: Yes

Reviewer #5: Yes

Reviewer #6: Yes

6. Review Comments to the Author

Reviewer #4: The authors have made a lot of efforts on refining this paper based on previous suggestions and now it looks very good.

I have no more questions except that could you provide more information about the intervention in this paper (how many questions solved by participants in the intervention group or ultimately which level did they achieve or frequently choose) to ensure that participants are indeed engaged in the intervention process? Because participants finished the intervention at home where they can be easily distracted, the performance on the calculation tasks may also be important indicators for engagement.

Reviewer #5: It is an interesting question to explore the effects of the abacus-based mental calculation training on cognitive function in older people. In the title, authors focused on the effects on cognitive function, but the results did not support the aim or the conclusion of the study. They only observed group differences in two tasks "Follow the order" after 2 months and "Route 99" after 6 months, and this may only be due to familiarity with digits/numbers with SoroTouc training, instead of improvement of working memory or attentional ability.

The second question is the sample size was two small, only 10 participants in each group.

it is unclear in the Fig.3, missing SD, line marker etc.

Reviewer #6: General Comments

Thanks for the invitation to review this article. It was a challenge to review the article after 3 other reviewers had previously reviewed it. The article has potential for publication, considering that it investigates the effect of an intervention with home-based computadorized cognitive function using mental calculation on the improvement/maintenance of cognitive function in healthy middle-aged and older people in a country where the number of older people has grown significantly.

Major comments

1. As this is a randomized clinical trial, I suggest a more detailed description of both groups at baseline (table 1). Below is an image that may help with this breakdown.

2. In the supplementary material “S2 table 2”, the “baseline scores” and “amount of change” data are exactly the same for the SoroTouch group. It is important to correct these values.

3. I believe that table 3 can be better explored. More information can be added, such as the mean total MOCA score for the control and intervention groups in the pre- and post-intervention period. For exemplo:

4. An alternative to statistical analysis can be the use of Generalized Estimating Equations (GEE). GEE have been repeatedly applied in controlled clinical trials. GEE and Bonferroni post hoc tests can be used for the comparison between moments (pre- and post-training) and groups.

5. Finally, I think it is important to included in the body and footnotes of the table must describe in detail the statistical method and results of the analysis.

7. PLOS authors have the option to publish the peer review history of their article (what does this mean?). If published, this will include your full peer review and any attached files.

Reviewer #4: No

Reviewer #5: No

Reviewer #6: **Yes: **César Augusto Häfele

---

## [Author Response · Author response to Decision Letter 2]

6 Oct 2023

We are very grateful for your careful review. Here are the responses to your comments. 

Reviewer #4: The authors have made a lot of efforts on refining this paper based on previous suggestions and now it looks very good.

I have no more questions except that could you provide more information about the intervention in this paper (how many questions solved by participants in the intervention group or ultimately which level did they achieve or frequently choose) to ensure that participants are indeed engaged in the intervention process? Because participants finished the intervention at home where they can be easily distracted, the performance on the calculation tasks may also be important indicators for engagement.

Response: We are grateful for your important suggestion. We agree with the suggestion that the participants' performance on the SoroTouch may have influenced changes in their cognitive functions. We added S1 Table” Performance during the intervention for each participant in the SoroTouch group.” The table includes the usage time, number of answers, number of correct answers, number of wrong answers, and final level of achievement.

Reviewer #5: It is an interesting question to explore the effects of the abacus-based mental calculation training on cognitive function in older people. In the title, authors focused on the effects on cognitive function, but the results did not support the aim or the conclusion of the study. They only observed group differences in two tasks "Follow the order" after 2 months and "Route 99" after 6 months, and this may only be due to familiarity with digits/numbers with SoroTouc training, instead of improvement of working memory or attentional ability.

Response: We agree with your opinion. It is possible that participants' number skills improved rather than their working memory, since "Route99" and "Follow the Order" were similar to SoroTouch in their use of numbers. We have added this description to the Discussion. (See p.19.)

The second question is the sample size was two small, only 10 participants in each group.

Response: We also think the small sample size is a limitation of this study. We have revised the relevant part of the Limitation. (See p. 20.)

it is unclear in the Fig.3, missing SD, line marker etc.

Response: Thank you for your appropriate comments. SE error bars added to Fig 3. SE is shown instead of SD because of the comparison of the population means between the two groups.

Reviewer #6: General Comments

Thanks for the invitation to review this article. It was a challenge to review the article after 3 other reviewers had previously reviewed it. The article has potential for publication, considering that it investigates the effect of an intervention with home-based computadorized cognitive function using mental calculation on the improvement/maintenance of cognitive function in healthy middle-aged and older people in a country where the number of older people has grown significantly.

Major comments

1. As this is a randomized clinical trial, I suggest a more detailed description of both groups at baseline (table 1). Below is an image that may help with this breakdown.

Response: We agree with your comments. 'BMI' and 'Education' in Table 1 were listed as continuous variables. (See p. 8.) As reviewer 3 pointed out as inappropriate at the first review, we did not apply statistical tests between the two groups for characteristics.

2. In the supplementary material “S2 table 2”, the “baseline scores” and “amount of change” data are exactly the same for the SoroTouch group. It is important to correct these values.

Response: We appreciate your suggestion. We have corrected that part.

3. I believe that table 3 can be better explored. More information can be added, such as the mean total MOCA score for the control and intervention groups in the pre- and post-intervention period. For exemplo:

Response: We have added the MoCA-J mean scores for both groups pre- and post-intervention in Table 3, following your suggestion. (See p. 16-17.)

4. An alternative to statistical analysis can be the use of Generalized Estimating Equations (GEE). GEE have been repeatedly applied in controlled clinical trials. GEE and Bonferroni post hoc tests can be used for the comparison between moments (pre- and post-training) and groups.

Response: We agree with your comments regarding GEE. However, we planned this trial with a mixed-effect model approach, so we analyzed data by MMRM. Additionally, we don't consider there are any drawbacks to analyzing the data with MMRM.

5. Finally, I think it is important to included in the body and footnotes of the table must describe in detail the statistical method and results of the analysis.

Response: As you indicated, we have described the statistical methods and results of the analysis in detail in the footnotes of the tables.

---

## [Decision Letter · Decision Letter 3]

18 Jan 2024

PONE-D-23-01459R3Effects of the abacus-based mental calculation training application "SoroTouch" on cognitive functions: a randomized controlled trialPLOS ONE

Dear Dr. Takaoka,

Thank you for submitting your manuscript to PLOS ONE. After careful consideration, we feel that it has merit but does not fully meet PLOS ONE’s publication criteria as it currently stands. Therefore, we invite you to submit a revised version of the manuscript that addresses the points raised during the review process. Please address comments raised by reviewer 6. 

We look forward to receiving your revised manuscript.

Kind regards,

Laura Kelly

Division Editor

PLOS ONE

Journal Requirements:

Reviewers' comments:

Reviewer's Responses to Questions

**Comments to the Author**

1. If the authors have adequately addressed your comments raised in a previous round of review and you feel that this manuscript is now acceptable for publication, you may indicate that here to bypass the “Comments to the Author” section, enter your conflict of interest statement in the “Confidential to Editor” section, and submit your "Accept" recommendation.

Reviewer #6: All comments have been addressed

2. Is the manuscript technically sound, and do the data support the conclusions?

Reviewer #6: Yes

3. Has the statistical analysis been performed appropriately and rigorously? 

Reviewer #6: Yes

4. Have the authors made all data underlying the findings in their manuscript fully available?

Reviewer #6: Yes

5. Is the manuscript presented in an intelligible fashion and written in standard English?

Reviewer #6: Yes

6. Review Comments to the Author

Reviewer #6: The authors carried out most of the recommendations. I think it is important to make the changes suggested below for publishing the article:

- In table 3, I suggest using “;” between the CI.

- In table 3, I suggest adding the CI to the pre-intervention values for both groups.

- I suggest again that the type of statistical test that was used for each variable be added to the footnotes of the tables.

7. PLOS authors have the option to publish the peer review history of their article (what does this mean?). If published, this will include your full peer review and any attached files.

Reviewer #6: No

---

## [Author Response · Author response to Decision Letter 3]

24 Jan 2024

We are very grateful for your careful review. Here are the responses to your comments. 

Journal Requirements:

Response: Thank you for your important suggestion. No papers were retracted. The information about the percentage of the population aged 65 and over in Reference No. 2, obtained from the Ministry of Internal Affairs and Communications of Japan's website, has been updated, and the text has been revised. (See p. 3) We also improved the style of the reference lists for Nos. 2 and 23.

Reviewer #6: The authors carried out most of the recommendations. I think it is important to make the changes suggested below for publishing the article:

- In table 3, I suggest using “;” between the CI.

Response: We appreciate your suggestion. We have corrected.

- In table 3, I suggest adding the CI to the pre-intervention values for both groups.

Response: Thanks for pointing this out. We have added 95% confidence intervals to the pre-intervention values for both groups in Table 3.

- I suggest again that the type of statistical test that was used for each variable be added to the footnotes of the tables.

Response: Thank you for your suggestion. We had described the statistical methods in the footnotes of the tables in response to your previous suggestion. If you have any further information that should be added, please let me know.

---

## [Editor Report · Decision Letter 4]

7 Feb 2024

Effects of the abacus-based mental calculation training application "SoroTouch" on cognitive functions: a randomized controlled trial

PONE-D-23-01459R4

Dear Dr. Takaoka,

We’re pleased to inform you that your manuscript has been judged scientifically suitable for publication and will be formally accepted for publication once it meets all outstanding technical requirements.

Kind regards,

Laura Kelly

Division Editor

PLOS ONE
---

## [Editor Report · Acceptance letter]

1 Mar 2024

PONE-D-23-01459R4 

PLOS ONE

Dear Dr. Takaoka, 

I'm pleased to inform you that your manuscript has been deemed suitable for publication in PLOS ONE. Congratulations! Your manuscript is now being handed over to our production team.

Kind regards, 

on behalf of

Dr. Laura Hannah Kelly 

Staff Editor

PLOS ONE